# Ectopic Eye Tooth Management: Photobiomodulation/Low-Level Laser Emission Role in Root Resorption after Fixed Orthodontic Treatment

**DOI:** 10.3390/healthcare10040610

**Published:** 2022-03-24

**Authors:** Mohammad Khursheed Alam, Kiran Kumar Ganji, Ahmed Ali Alfawzan, Srinivas Munisekhar Manay, Kumar Chandan Srivastava, Prabhat Kumar Chaudhari, Hala A. Hosni, Haytham Jamil Alswairki, Reem Ahmed Alansari

**Affiliations:** 1Orthodontics, Preventive Dentistry Department, College of Dentistry, Jouf University, Sakaka 72345, Saudi Arabia; 2Preventive Dentistry Department, College of Dentistry, Jouf University, Sakaka 72345, Saudi Arabia; dr.srinivas.manay@jodent.org; 3Department of Preventive Dentistry, College of Dentistry in Ar Rass, Qassim University, Ar Rass 52571, Saudi Arabia; ah.alfawzan@qu.edu.sa; 4Department of Oral & Maxillofacial Surgery & Diagnostic Sciences, College of Dentistry, Jouf University, Sakaka 72345, Saudi Arabia; drkcs.omr@gmail.com (K.C.S.); dr.hala.hosni@jodent.org (H.A.H.); 5Division of Orthodontics and Dentofacial Deformities, Centre for Dental Education and Research, All India Institute of Medical Sciences, New Delhi 110029, India; dr.prabhatkc@gmail.com; 6School of Dental Sciences, Universiti Sains Malaysia, Kota Bharu 16150, Malaysia; hitham.swerki@gmail.com; 7Department of Orthodontics, Faculty of Dentistry, King Abdulaziz University, Jeddah 22252, Saudi Arabia; ralansari@kau.edu.sa

**Keywords:** photobiomodulation, low level laser therapy, ectopic eye tooth, root resorption, treatment modalities

## Abstract

Aim: This study evaluates the role of low-level laser emission/photobiomodulation (LE/P) in quantitative measurements of root resorption (QRR). The application of LE/P performed after each orthodontic activation with four types of treatment intervention (TI) on the root resorption (RR) after fixed orthodontic treatment (FOT) of the upper arch with ectopic eye tooth/teeth [EET] was investigated. Materials and Methods: Thirty-two orthodontic patients scheduled for FOT were selected and assigned to the four groups. These were LE/P + Self ligating bracket (SLB), LE/P + Conventional bracket (CB), non-photobiomodulation (non-LE/P) + SLB and non-LE/P + CB. Standard management stages of FOT were followed in the maxilla. Each patient received a single application of LE/P labially/buccally and palatally, a total of five different points were used during each activation or appointment. The main outcome measure was QRR in maxillary anteriors before and after FOT, assessed via cone-beam computed tomography (CBCT) using 3D OnDemand software. Results: Insignificant QRR was found between before and after FOT in SLB, CLB, and LE/P, non-LE/P groups (*p* > 0.05). QRR in the SLB vs. CB and LE/P vs. non-LE/P group was significantly different in 11, 13, and 23 (*p* < 0.05). QRR in the LE/P + SLB group (*p* < 0.05) was significantly different in 11, 13, and 23 than that in the other groups. The most severe QRR was found on 13 (0.88 ± 0.28 mm and 0.87 ± 0.27 mm) and 23 (1.19 ± 0.14 mm and 1.16±0.13 mm) in the CB and non-LE/P group (*p* < 0.001). LE/P + SLB showed a highly significant superior outcome (*p* < 0.001) in relation to non-LE/P + CB, the QRR of 23 were 0.813 ± 0.114 mm and 1.156 ± 0.166 mm, respectively. Conclusion: Significantly higher amounts of QRR were found in EET patients after FOT treated with the CB, non-LE/P, and non-LE/P + CB system and warrant further investigation to explore potential specific causes.

## 1. Introduction

The primary concern of any orthodontic management is the improvement of dental and facial aesthetics rather than other oral health benefits in patients [1,2]. Every intervention has risk or impediments, fixed orthodontic treatment (FOT) is not free from adverse effects. For orthodontic tooth movement (OTM), the unbalanced force might result in detrimental treatment consequences like root resorption (RR), pain, overdue tooth movement, and loss of vitality of the tooth [3]. Different studies established that OTM is a complex phenomenon and involves loss of the alveolar bone or tooth root [4,5].

Malocclusion is a common phenomenon, which can be seen in a large portion of the individuals around the globe. Malocclusion is considered as off base connection between and among the misalignment of the teeth of both arches [1,6,7]. This condition, as a rule, could be perceived at an early age and becomes apparent bit by bit with development and subsequently informs patients whether they should seek out FOT.

An ectopic eye tooth (EET) is a maxillary canine which follows a deviant route of eruption. Due to such barriers, EET can be impacted or blocked from erupting or displaced buccally or palatally. Regardless of advancements in the field of orthodontics, fruitful management of EET still represents a challenge to clinicians. They are normally guided to an appropriate arrangement in the dental arch by surgical exposure followed by orthodontic traction [8].

RR is one of the most widely recognized unfavorable impacts of a wide range of orthodontic treatments, which may begin in any phase of the treatment, for example, starting, retraction, alignment, and completing stage. RR may initiate around 2–5 weeks of activation, and it requires 3–4 such activations to be recognized in radiographs [9]. Although many aspects of this unfortunate RR effect are unknown, it is a perplexing natural process that occurs when orthodontic activation at the root apex region exceeds the opposition and reparative capacity of the periapical tissues [10]. The RR during orthodontic therapy is likely to be influenced by environmental and hereditary factors [11].

Photobiomodulation (LE/P) has promising advantages on OTM [12,13], pain management [12,14] and gingival tissue and periodontal tissue. It has been uncovered that LE/P essentially increment osteoblast cells during their expansion and separation stages. In this manner, it prompts bone renovation by invigorating osteogenesis [15,16]. Quality bone recovery prompts the better bone redesigning, which is basic for OTM. To date, based on a literature search, no study has been found which assessed QRR associated with LE/P in FOT. Hence, the general research objectives were to explore QRR via 3D CBCT after FOT in the patients undergoing fixed orthodontic treatment of maxillary arch with ectopically erupted canines. Specifically, the current study compares the QRR

in two-treatment modalities namely self-ligating bracket (SLB) vs. conventional bracket (CB).in two-treatment intervention (TI) (LE/P vs. non-LE/P).in four different TI (LE/P + SLB, LE/P + CB, non-LE/P + SLB, and non-LE/P + CB).in three different phenotypes of EET (Unilateral 13, Unilateral 23 and Bilateral)in three different phenotype positions of EET (Buccal, Occlusal and Palatal)

**Hypothesis** **1.***There was a significant difference of the QRR in different types of fixed appliances, different treatment interventions and different phenotypes of ectopically erupted canines*.

## 2. Materials and Methods

Thirty-two (32) patients were enrolled and were further subdivided into two groups based upon the two different types of orthodontic appliances used, i.e., SLB and CB. These two groups were further subdivided into two subgroups based on the TI using LE/P. The different study groups have been presented in Figure 1. Inclusion criteria: Patients with an age group of 14–25 diagnosed as Angle Class I or II or III malocclusion with an ectopic maxillary canine, requiring space creation or extraction of first premolar. Patients on long-term medication, craniofacial anomalies/malformation, with parafunctional habits, temporomandibular joint dysfunction, multiple missing teeth, and periodontally compromised were excluded from the study. G*Power software version 3.0.10 (Franz Faul Universitat, Kiel, Germany) with power 80%, α 0.05 and effect size (d) 0.22 was used. Hence, the total sample size intended for this research was 32, each group required a minimum of eight subjects.

The detailed method for QRR measurements outcome [17] and its reliability are presented in Figure 2. CBCT were taken during initial record (Pre) and the day of debonding (Post FOT), which were used for the RR outcome measurements. The average duration of FOT was 19.40 and 20.63 months in the SLB and CB group, respectively.

### Statistical Analysis

A paired *t*-test, independent *t*-test and ANOVA with post-hoc Tukey tests are used for testing the comparisons in groups. SPSS version 26 (Chicago, IL, USA) is used for the analysis.

## 3. Results

The average age of the patients was 19.5 (±1.5) with 18 males and 14 females. A paired-sample *t*-test was conducted to evaluate the impact of the SLB and CB on the level of QRR after FOT. The results showed an insignificant (*p* > 0.05) QRR in both the groups (Table 1). The mean increase in QRR was highest in 23 of both the techniques SLB and CB with a 95% CI ranging from 0.94 to 1.19. The effect size static (0.97) was highest in relation to #23 of the CB group.

The results of independent *t*-test performed for the mean QRR (#11, #12, #21, #22, #13 and #23) in SLB vs. CB groups revealed statistically significant disparities only in #11, #13, and #23 (*p* < 0.05) (Figure 3).

A paired-samples *t*-test was conducted to evaluate the impact of the LE/P and non-LE/P on the level of QRR. The results showed an insignificant (*p* > 0.05) QRR in both the groups after FOT (Table 2). The mean increase in QRR was highest in #23 of both the techniques LE/P and non-LE/P with a 95% CI ranging from 0.93 to 1.16. The effect size static (0.92) was highest in relation to #23 of non-LE/P, indicating a high rate of resorption in non-LE/P group.

The results of an independent *t*-test performed for the mean QRR (#11, #12, #21, #22, #13, and #23) in LE/P and non-LE/P groups revealed statistically significant disparities in #11, #13, and #23 (*p* < 0.05) (Figure 4).

The type of technique (LE/P and non-LE/P) and type of brackets (SLB/CB) had a significant impact on the resorption levels in #11 (F = 3.830), #13 (F = 5.076) and #23 (F = 15.521) *p* < 0.05, respectively (Table 3).

Graph 3 revealed that there was a significant difference between the mean resorption levels of #11 in LE/P + SLB and LE/P + CB with that of non-LE/P + CB. With respect to #13, there was significant difference between the mean resorption levels in LE/P + SLB and non-LE/P + SLB with that of non-LE/P + CB, whereas in the case of #23, a significant difference was found in LE/P + SLB with that of non-LE/P + SLB, LE/P + CB and non-LE/P + CB.

The ectopic canine pattern and canine position did not show any significant impact on the resorption levels of #11, #21, #12, #22, #13, and #23 (Table 4 and Table 5, Figure 5).

## 4. Discussion

RR negatively affects patients’ quality of life and treatment result during FOT. LE/P uses are popular in FOT and have been successfully tested in animal [15,16] and human studies where pain management [12,13,14] and acceleration of OTM [12,13] are common. However, QRR during the FOT using LE/P is yet to explored. The current study explored and compared the QRR in FOT patients treated with different TM (two groupings), TI (two and four groupings) and different phenotype factors (two different three groupings) of EET using CBCT, and found significant differences.

Treatment of EET is a costly, tedious procedure by fixed braces and requires a measure of time contingent on the case to adjust them inside the dental arch [8]. The orthodontic sections can then be attached to the uncovered canines (in cases where they are impacted) and adjusted [8]. RR is where the loss of dental hard tissues is happening because of the clastic action of bone [18]. This phenomenon may likewise result in an obsessive or physiological procedure. Be that as it may, RR is considered to be an ordinary physiological procedure in deciduous teeth except if it happens prematurely [19]. Numerous examinations have shown that RR is unequivocally affected by several environmental factors. Different mechanical variables may likewise be included in RR during FOT, for example, rotation, supraocclussion, root tipping and infraocclusion, etc. These relevant factors generally used during the OTM stage or completing phase of FOT, which may have exaggerated the formation of RR. Consequently, to recognize the impact of the treatment variety and mechanical consequences for RR, they ought to be directed at a phase during FOT. Maybe, toward the end of FOT, a system could be determined that led to the progression of RR, for example, the use of excessive force mechanics.

To investigate RR outcomes, cephalogram, panoramic, and peri-apical radiographs were common using different techniques of measurement [20,21,22]. Present research used CBCT images to measure QRR outcomes. CBCT images have a high level of reproducibility and confirm its usage in FOT [23,24]. Alqerban et al. (2009) and Li et al. (2020) also used the CBCT for the QRR outcome measurements [17,23]. The outputs created by a CBCT imaging framework with an altogether decreased sweep time diminished expense for the patient, and utilized a lower radiation portion than computed tomography (CT) [25].

Several investigations advocate for SLB vs. CB in FOT [26,27,28], with an insignificant QRR outcome. Most of them investigate the outcome in the maxillary arch or only in incisor teeth. The current study assessed all anterior teeth of maxilla in a prospective study design, others were controlled clinical trials or cohort studies. Li et al. (2020) analyzed the prevalence and severity of QRR in the clear aligners and FOT group were statistically and clinically significant [17]. Treatment groupings are completely different. In relation to FOT (average 1.12 ± 1.34 mm), clear aligners (average 0.13 ± 0.47 mm) were used in relatively simpler cases where QRR in all anterior teeth were significantly less. In the FOT group, Li et al. (2020) found QRR differences (before and after) in the maxillary central incisor (1.23 ± 1.31 mm), lateral incisor (1.31 ± 1.33 mm), and canine (1.53 ± 1.92 mm) [17]. In the current study, FOT using CB show the QRR differences in 11 (0.203 mm), 21 (0.233 mm), 12 (0.176), 22 (0.277 mm), 13 (0.878 mm), 23 (1.189 mm), and SLB show 11 (0.296 mm), 21 (0.228 mm), 12 (0.181), 22 (0.292 mm), 13 (0.572 mm), 23 (0.939 mm). In comparison to TM, 11, 13 and 23 QRR were significantly different. These results coincide with the findings of Aras et al. (2018). There were differences in grouping, patients’ selection, and the measurement timings [29]. However, the number of subjects, and the TM protocol were same. Aras et al. (2018), investigated 32 subjects with Class I malocclusion having crowding level 4–10 mm and QRR compared between SLB and CB in CBCT [29]. Nine months after FOT, they found higher incidence of slanted QRR with the CB system [29]. The current study also found differences in QRR in CB and the changes investigated after FOT.

The effectiveness of LE/P using the TM, TI, and different phenotype factors of EET were observed on the maxillary anterior tooth QRR, and LE/P was applied at 4 week intervals, which is common protocol to call FOT patients in at regular intervals for activation/adjustment.

The current study showed in comparison to TI (LE/P vs. non-LE/P), 11, 13 and 23 QRR were significantly different. Furthermore, in four different TI groups there were significant difference in QRR. LE/P + SLB showed significant differences in QRR of 11 (vs. LE/P + CB and non-LE/P + CB), 13 (vs. non-LE/P + CB) and 23 (with all groups). Based on the results, LE/P + SLB reflects the superiority in the QRR outcome. However, these aftereffects are incomparable as, to date, no research has been explored investigating the QRR association with LE/P vs. non-LE/P in FOT and LE/P + SLB vs. non-LE/P + SLB vs. LE/P + CB vs. non-LE/P + CB. Ng et al., 2017 and Khaw et al. (2018), investigated RR after administration of LE/P determining the aftereffects on extracted premolars [30,31]. Ng et al. (2017) found LE/P has a better outcome in RR compared to control [30]. Moreover, Khaw et al. (2018) investigated the effect of LE/P on the healing of RR on extracted teeth and an excellent outcome has been observed [31].

The outcome of QRR in two different phenotype factors of EET were insignificant. Unilateral right or left and bilateral EET and buccally, occlusal and palatally placed EET had insignificant changes in QRR outcomes. Brusveen et al. (2012) investigated QRR outcome using intra-oral radiographs of before and after FOT in unilateral EET vs. control and found insignificant changes [22]. In a different protocol Lempesi et al. (2014) compared the outcome of QRR in 24 unilateral/bilateral EET vs. 24 controls and found no differences [32].

RR is quite common in FOT. Several TI and TM were investigated by researchers to lessen the RR [33]. Genetics, phenotype, geomorphometry, and type of TI and TM may play an important role in the outcome of RR [33,34]. This first-in-human study in the Saudi population in relation to 2 TI and 2 + 4 TM and in different phenotype factors of EET after FOT explored the amount QRR. This study was a prospective study and used CBCT for QRR outcome measurements with an appropriate sample size; however, investigation with a larger sample can certainly give better outcomes. Additionally, this study is single center based, a multiple center-based study might explore more certain conclusions. Knowing the extent of EET patient after FOT that are associated with QRR before starting any TI and TM, and being able to discuss such information with the patient, parents, and guardians, allows the choice of the most suitable FOT. Lastly, a future investigation after long term follow-up in the retention phase and relapse level might give different results in the outcome of QRR measurements.

## 5. Conclusions

QRR on CBCT in EET patients after FOT with SLB, LE/P, and LE/P + SLB showed a superior outcome. It is difficult to affirm superiority of one TM or TI over the other only considering QRR amounts which affirms more research utilizing similar protocol in different centers to identify possible similarities and disparities to accept the TM and TI.

Based on the results:

Objectives 1—in two treatment modalities (TM) [SLB vs. CB]—significant disparities are found, SLB having a superior outcome measure of QRR after FOT.

Objectives 2—in two treatment interventions (TI) [LE/P vs. non-LE/P]—significant disparities are found, LE/P having superior outcome measure of QRR after FOT.

Objectives 3—in four different TI (LE/P + SLB, LE/P + CB, non-LE/P + SLB, and non-LE/P + CB)—significant disparities are found, LE/P + SLB having superior outcome measure of QRR after FOT.

Objectives 4 and 5—in three different phenotypes of EET (Unilateral 13, Unilateral 23 and Bilateral) and positions of EET (Buccal, Occlusal and Palatal)—insignificant disparities were observed.

## Figures and Tables

**Figure 1 healthcare-10-00610-f001:**
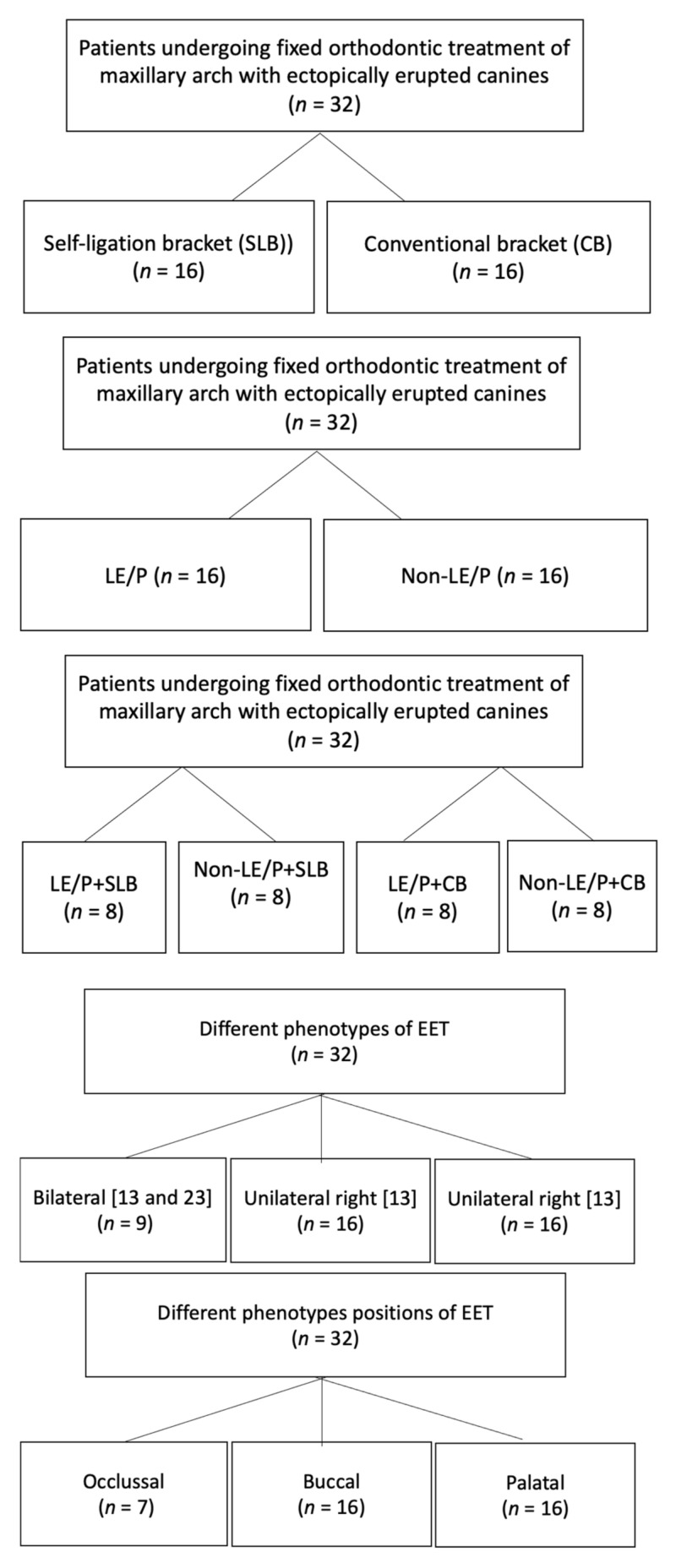
Group allocation of study population.

**Figure 2 healthcare-10-00610-f002:**
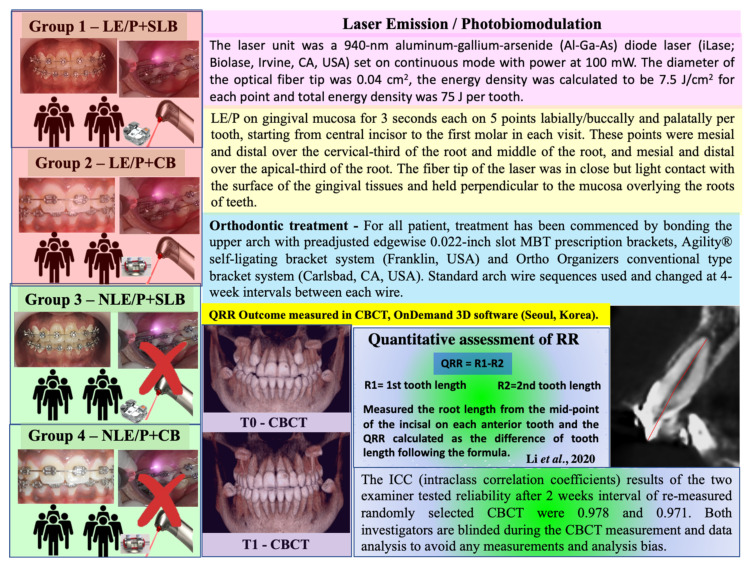
Details of photobiomodulation (LE/P) and quantitative assessment of RR.

**Figure 3 healthcare-10-00610-f003:**
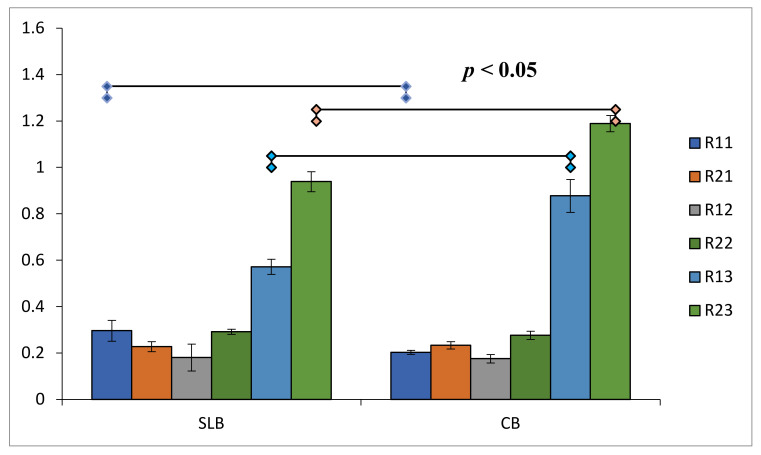
Graphical representation of significance difference in root resorption within intra-group comparison of SLB and CB.

**Figure 4 healthcare-10-00610-f004:**
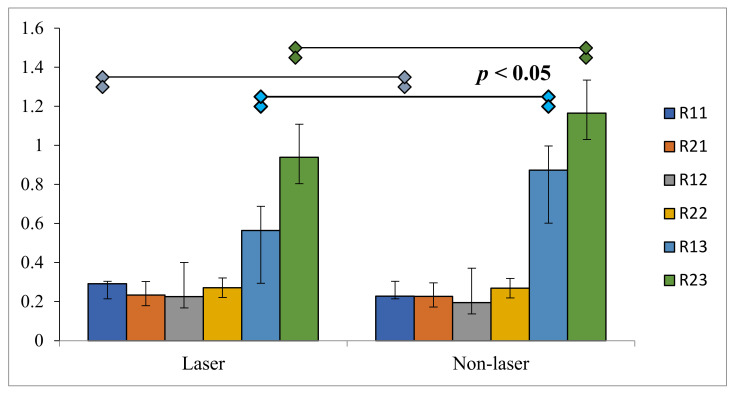
Graphical representation of significance differences in root resorption within intra-group comparison of laser and non-laser groups.

**Figure 5 healthcare-10-00610-f005:**
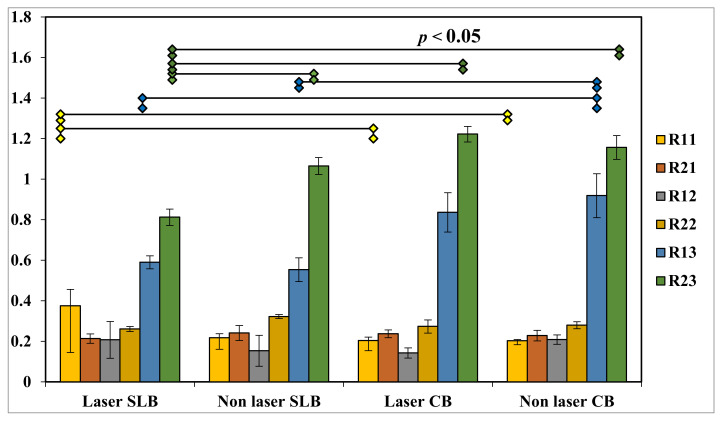
Graphical representation of Tukey’s post-hoc test results for intra-group significance.

**Table 1 healthcare-10-00610-t001:** Mean comparison of root resorption after orthodontic therapy in SLB and CB group.

Root Length Comparison before (R1) and after (R2) Orthodontic Treatment	Group	N	Mean Root Resorption R = (R2 − R1) mm	Std. Deviation	95% Confidence Interval of the Difference		t-Value
					Lower	Upper	
11R1−11R2	SLB	16	0.29	0.18	0.19988	0.39262	2.02
	CB	16	0.20	0.04	0.18389	0.22236	
21R1−21R2	SLB	16	0.23	0.08	0.18226	0.27274	−0.21
	CB	16	0.23	0.06	0.19955	0.26670	
12R1−12R2	SLB	16	0.18	0.23	0.05793	0.30332	0.08
	CB	16	0.18	0.07	0.13628	0.21497	
22R1−22R2	SLB	16	0.29	0.04	0.26827	0.31548	0.71
	CB	16	0.28	0.07	0.23916	0.31459	
13R1−13R2	SLB	16	0.57	0.13	0.50228	0.64147	−3.91
	CB	16	0.88	0.28	0.72614	1.02886	
23R1−23R2	SLB	16	0.94	0.17	0.84683	1.03067	−4.51
	CB	16	1.19	0.14	1.11499	1.26376	

**Table 2 healthcare-10-00610-t002:** Mean comparison of root resorption after orthodontic therapy in laser and non-laser groups.

Root Length Comparison before (R1) and after (R2) Orthodontic Treatment	Group	N	Mean Root Resorption R = (R2 − R1) mm	Std. Deviation	95% Confidence Interval of the Difference		t-Value
					Lower	Upper	
11R1−11R2	LE/P	16	0.30	0.08	0.01830	0.10920	0.28
	Non-LE/P	16	0.22	0.05	0.01797	0.10953	
21R1−21R2	LE/P	16	0.23	0.07	−0.03820	0.05195	0.31
	Non-LE/P	16	0.23	0.05	−0.03833	0.05208	
12R1−12R2	LE/P	16	0.23	0.18	−0.06414	0.12539	0.66
	Non-LE/P	16	0.19	0.06	−0.06681	0.12806	
22R1−22R2	LE/P	16	0.27	0.05	−0.03439	0.03814	0.10
	Non-LE/P	16	0.27	0.05	−0.03439	0.03814	
13R1−13R2	LE/P	16	0.56	0.12	−0.46058	−0.15692	−4.15
	Non-LE/P	16	0.87	0.27	−0.46331	−0.15419	
23R1−23R2	LE/P	16	0.93	0.17	−0.33628	−0.11497	−4.16
	Non-LE/P	16	1.16	0.13	−0.33653	−0.11472	

**Table 3 healthcare-10-00610-t003:** ANOVA test results of mean root resorption (R) for the measured tooth (#11, #21, #12, #22, #13, #23) within laser SLB, non-laser SLB, laser CB and non-laser CB group.

Tooth	Groups	N (32)	Mean Resorption (R)	Std. Deviation	F Value	*p* Value
11	LE/P + SLB	8	0.3750	0.22960	3.830	0.020 *
	Non-LE/P + SLB	8	0.2175	0.05651		
	LE/P + CB	8	0.2038	0.04955		
	Non-LE/P + CB	8	0.2025	0.01832		
21	LE/P + SLB	8	0.2137	0.06523	0.204	0.893
	Non-LE/P + SLB	8	0.2413	0.10371		
	LE/P + CB	8	0.2375	0.05548		
	Non-LE/P + CB	8	0.2287	0.07338		
12	LE/P + SLB	8	0.2075	0.25645	0.323	0.809
	Non-LE/P + SLB	8	0.1537	0.21494		
	LE/P + CB	8	0.1425	0.07086		
	Non-LE/P + CB	8	0.2088	0.06446		
22	LE/P + SLB	8	0.2612	0.03563	1.770	0.176
	Non-LE/P + SLB	8	0.3225	0.02816		
	LE/P + CB	8	0.2738	0.09164		
	Non-LE/P + CB	8	0.2800	0.04811		
13	LE/P + SLB	8	0.5900	0.09118	5.076	0.006 *
	Non-LE/P + SLB	8	0.5537	0.16578		
	LE/P + CB	8	0.8362	0.27412		
	Non-LE/P + CB	8	0.9187	0.30638		
23	LE/P + SLB	8	0.8125	0.11449	15.521	0.000 *
	Non-LE/P + SLB	8	1.0650	0.11928		
	LE/P + CB	8	1.2225	0.10859		
	Non-LE/P + CB	8	1.1563	0.16570		

* = *p* < 0.05.

**Table 4 healthcare-10-00610-t004:** ANOVA test results of mean root resorption (R) for the measured tooth (#11, #21, #12, #22, #13, #23) within bilateral, unilateral 13, unilateral 23 ectopic canine groups.

Tooth	Ectopic Canine	N(32)	Mean	Std. Deviation	F Value	*p* Value
11	Bilateral	9	0.1989	0.04343	1.703	0.200
	Unilateral 13	12	0.3033	0.20299		
	Unilateral 23	11	0.2327	0.07086		
21	Bilateral	9	0.2278	0.07259	0.007	0.993
	Unilateral 13	12	0.2308	0.04522		
	Unilateral 23	11	0.2318	0.10157		
12	Bilateral	9	0.2322	0.21730	0.636	0.536
	Unilateral 13	12	0.1600	0.12649		
	Unilateral 23	11	0.1536	0.16931		
22	Bilateral	9	0.2878	0.03701	1.002	0.380
	Unilateral 13	12	0.2667	0.05228		
	Unilateral 23	11	0.3009	0.07648		
13	Bilateral	9	0.6200	0.30749	0.975	0.389
	Unilateral 13	12	0.7750	0.29950		
	Unilateral 23	11	0.7555	0.18190		
23	Bilateral	9	1.0511	0.18937	0.236	0.791
	Unilateral 13	12	1.0425	0.18582		
	Unilateral 23	11	1.0982	0.23549		

**Table 5 healthcare-10-00610-t005:** ANOVA test results of mean root resorption for the measured tooth (#11, #21, #12, #22, #13, #23) within occlusal, buccal, palatal canine position groups.

Tooth	Canine Position	N (32)	Mean	Std. Deviation	F Value	*p* Value
11	Occlusal	7	0.2600	0.07000	0.964	0.393
	Buccal	13	0.2108	0.03883		
	Palatal	12	0.2858	0.21232		
21	Occlusal	7	0.2500	0.07047	0.346	0.710
	Buccal	13	0.2208	0.06551		
	Palatal	12	0.2292	0.08670		
12	Occlusal	7	0.1986	0.19239	0.158	0.855
	Buccal	13	0.1869	0.14338		
	Palatal	12	0.1567	0.19047		
22	Occlusal	7	0.2657	0.04467	1.095	0.348
	Buccal	13	0.2769	0.06316		
	Palatal	12	0.3033	0.05959		
13	Occlusal	7	0.6243	0.19139	0.772	0.472
	Buccal	13	0.7246	0.24939		
	Palatal	12	0.7833	0.32129		
23	Occlusal	7	1.1286	0.21575	0.621	0.545
	Buccal	13	1.0685	0.18920		
	Palatal	12	1.0217	0.20919		

## Data Availability

The data used to support the findings of this study are included in the article.

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
