# Peer review of "Ectopic Eye Tooth Management: Photobiomodulation/Low-Level Laser Emission Role in Root Resorption after Fixed Orthodontic Treatment"

_healthcare, 2022, doi:10.3390/healthcare10040610_

Round 1

Reviewer 1 Report

Dear Authors

the article is interesting but it needs some changes before taking it into consideration for publication. Attached you can find my suggestions.

According to the analysis of the scientific article: Ectopic eye tooth management: Photobiomodulation / low-level laser emission role in root resorption after fixed Orthodontic  treatment.”  the following observations were made:

ABSTRACT 

  • The abstract is very extensive and does not describe the total content of the article and for this reason it should be more synthetic, eliminating unnecessary parts such as according to the text: " The orthodontic treatment brings numerous benefits and, in most cases, the benefits outweigh the possible disadvantages. Root resorption (RR) is a common adverse phenomenon  associated with orthodontic treatment. "
  • The abstract being very extensive, for a better understanding of the content, the subtitles should be considered: aim, materials and methods, results and conclusion.
  • The abstract does not present a conclusion.
  • In the aim of this study, it does not determine the specific study variable, such as the presence or absence of root resorption or the longitudinal measurement of resorption present before and after orthodontic treatment.

INTRODUCTION

  • According to the article Hypotheses There was a significant difference in the QRR in different types of fixed appliances, different treatment interventions and different phenotype of ectopically erupted canines.  Research Question  In the patients undergoing FOT of maxillary arch with EET (unilateral or bilateral) with  different phenotype positions (buccal, occlusal, palatal) using SLB and CB system, what 95 is the effect of LE/P intervention as compared to control on the QRR? “ but is it necessary to consider this text in the introduction of a scientific article?

MATERIALS AND METHODS

  • According to figure 1 of the present study, it considers the formation of 4 study groups according to 5 independent variables (type of bracket, different phenotypes of EET, different phenotypes, position of EET and the presence or absence of laser use) but as observed in figure 1 these groups were not formed considering in each group the presence of these 5 independent variables and for this reason it is considered that the groups formed of 8 people each were not correct.
  • According to the article “ LE/P on gingival mucosa for 3 seconds each on 5 points labially/buccally and palatally per tooth, starting from central incisor to the first molar in each visit. These points were mesial and distal over the cervical-third of the root and middle of the root, and mesial and distal over the apical-third of the root. “ but the figure 1 mentions the presence of 3 different position phenotypes and the teeth evaluated were from 13 to 23 (as evidenced in tables 1 to 5).
  • Considering 8 people in each group is a very small sample

RESULTS

  • According to the article “ The type of technique (LE/P and non-LE/P) and type of brackets (SLB/CB) had a signifcant impact on the resorption levels in #11 (F=3.830), #13 (F= 1.770) and #23 (F=15.521) p<0.05 respectively (Table 3). “ but according to table 3 the value of F for #13 is 5.076 and not 1.770.
  • The content of the results obtained in different tables and figures of this article are mostly descriptive and not analytical.

Best regards 

Author Response

Dear Authors

the article is interesting but it needs some changes before taking it into consideration for publication. Attached you can find my suggestions.

Answer: Respected Sir, thank you very much for your kind help. We did the changes as advised. Looking forward to your kind help.

According to the analysis of the scientific article: “Ectopic eye tooth management: Photobiomodulation / low-level laser emission role in root resorption after fixed Orthodontic treatment.”  the following observations were made:

ABSTRACT 

  • The abstract is very extensive and does not describe the total content of the article and for this reason it should be more synthetic, eliminating unnecessary parts such as according to the text: " The orthodontic treatment brings numerous benefits and, in most cases, the benefits outweigh the possible disadvantages. Root resorption (RR) is a common adverse phenomenon associated with orthodontic treatment. "

Answer: The abstract is made to be very much precise by eliminating the text from it.  Please refer line 24-26.

  • The abstract being very extensive, for a better understanding of the content, the subtitles should be considered: aim, materials and methods, results and conclusion.

Answer: The abstract has been revised to be as a structured abstract for better understanding of the content.

  • The abstract does not present a conclusion.

Answer: The conclusion has been provided in the abstract.

  • In the aim of this study, it does not determine the specific study variable, such as the presence or absence of root resorption or the longitudinal measurement of resorption present before and after orthodontic treatment.

Answer: We understand the concern of the reviewer. The general research objectives was to explore QRR via 3D CBCT after FOT in the patients undergoing fixed orthodontic treatment of maxillary arch with ectopically erupted canines. QRR was the dependent variable which was assessed before and after orthodontic therapy as the same evaluation was done with various treatment interventions. Please refer line 83-92.

INTRODUCTION

  • According to the article “ Hypotheses There was a significant difference in the QRR in different types of fixed appliances, different treatment interventions and different phenotype of ectopically erupted canines.  Research Question  In the patients undergoing FOT of maxillary arch with EET (unilateral or bilateral) with  different phenotype positions (buccal, occlusal, palatal) using SLB and CB system, what 95 is the effect of LE/P intervention as compared to control on the QRR? “but is it necessary to consider this text in the introduction of a scientific article?

Answer: First of all, thank you for your constructive comments and suggestions that allowed us to greatly improve the quality of the manuscript. As requested by the reviewer the research question is being deleted from the introduction section. Only hypothesis is retained so that readers can catch the point of interest in easier terms.

MATERIALS AND METHODS

  • According to figure 1 of the present study, it considers the formation of 4 study groups according to 5 independent variables (type of bracket, different phenotypes of EET, different phenotypes, position of EET and the presence or absence of laser use) but as observed in figure 1 these groups were not formed considering in each group the presence of these 5 independent variables and for this reason it is considered that the groups formed of 8 people each were not correct.

Answer: We thank the reviewer for his useful insight into the descriptive aspect of the methodology. The grouping system in Figure 1 is being revised to be more clear and self-explanatory.

  • According to the article “ LE/P on gingival mucosa for 3 seconds each on 5 points labially/buccally and palatally per tooth, starting from central incisor to the first molar in each visit. These points were mesial and distal over the cervical-third of the root and middle of the root, and mesial and distal over the apical-third of the root. “ but figure 1 mentions the presence of 3 different position phenotypes, and the teeth evaluated were from 13 to 23 (as evidenced in tables 1 to 5).

Answer: We thank and appreciate the reviewer for pointing out the error in methodology. Initially, data collection was done for the teeth from the central incisor till the first molar of each quadrant as part of the comprehensive effect of laser therapy. As per the objective of the study, we intended to present the data related to 13 to 23 only, hence the changes in the methodology section have been updated to present that LE/P was done from 13 to 23.

  • Considering 8 people in each group is a very small sample

Answer: The sample size was estimated using G power computing tool version 3.0.10. With the power of 80% and α 0.05 and effect size (d) as .22. The total sample size estimated for the study was 32 hence each group had 8 subjects.

RESULTS

  • According to the article “ The type of technique (LE/P and non-LE/P) and type of brackets (SLB/CB) had a significant impact on the resorption levels in #11 (F=3.830), #13 (F= 1.770) and #23 (F=15.521) p<0.05 respectively (Table 3). “ but according to table 3 the value of F for #13 is 5.076 and not 1.770.

Answer: We apologize for the typing error caused.  The F values related to #13 are being updated correctly to match with that of Table 3.

  • The content of the results obtained in different tables and figures of this article are mostly descriptive and not analytical.

Answer: We thank the reviewer for his valuable input. Yes, the description of results was limited to be more descriptive as it avoids repetition of data being presented. The precision of presenting results was adopted to address each objective in detail and every attempt is made to deviate from the said objectives.

Reviewer 2 Report

Line 56 and line 191: marvel? What the authors mean by using the term marvel, which to my knowledge refers more to a prodigy, something absolutely extraordinary...

Line 58 malalignment? change with misalignment

Line 70 e 71 Albeit numerous parts of this unfortunate impact stay hazy? It is necessary to better define what the authors intend to describe

Line 73 The hereditary impact? it is necessary to better define what the authors intend to describe

Line 147 Figure 1. Details of the methodology:  Maybe Figure 2 Details of the methodology?

Figure 2: What it mean “Patient on long-term medication”, please better describe

I do not agree with having included the description of the criteria of inclusion/ exclusion, sample size calculation, age of subjects, laser emission / Photobiomodulation and the type of orthodontic treatment and also quantitative assessment of RR inside a figure….I suggest to make a separate paragraph to include this informations, which can then be summarised in one or more figures if the authors consider it necessary.

Minore language revison required

There are 13 self-citations out of a total of 42 references for the 1st author Mohammad Khursheed Alam, it seems too many self-citations.  I suggest removing the self-citations. 

Author Response

Answer: Respected Sir, thank you very much for your kind help. We did the changes as advised. Looking forward to your kind help.

Line 56 and line 191: marvel? What the authors mean by using the term marvel, which to my knowledge refers more to a prodigy, something absolutely extraordinary...

Answer: We regret for the misunderstanding caused. The sentence in line 51 & 191 has been rephrased to be easier to understand.

Line 58 malalignment? change with misalignment

Answer: Thanks for your valuable comment. The word has been replaced with misalignment.

Line 70 e 71 Albeit numerous parts of this unfortunate impact stay hazy? It is necessary to better define what the authors intend to describe

Answer: We apologize the error caused the sentence has been completely rephrased to better describe and define the intended concept about root resorption during orthodontic therapy. Please refer line 70-73.

Line 73 The hereditary impact? it is necessary to better define what the authors intend to describe

Answer: Thanks for your valuable input. The impact of hereditary on root resorption during orthodontic therapy is being explained clearly. Please refer line 73 -74.

Line 147 Figure 1. Details of the methodology:  Maybe Figure 2 Details of the methodology?

Answer: Yes the legend of Figure 2 has been that represent the  details of the methodology.

Figure 2: What it mean “Patient on long-term medication”, please better describe

Answer: Patients undergoing long-term therapy such as immunomodulatory drugs, immunosuppressant drugs, and NSAIDs were excluded from the study as these drugs influence the orthodontic tooth movement, (Ref: Diravidamani K, Sivalingam SK, Agarwal V. Drugs influencing orthodontic tooth movement: An overall review. J Pharm Bioallied Sci. 2012 Aug;4(Suppl 2):S299-303. doi: 10.4103/0975-7406.100278. PMID: 23066275; PMCID: PMC3467877.).

I do not agree with having included the description of the criteria of inclusion/ exclusion, sample size calculation, age of subjects, laser emission / Photobiomodulation and the type of orthodontic treatment and also quantitative assessment of RR inside a figure….I suggest to make a separate paragraph to include this informations, which can then be summarised in one or more figures if the authors consider it necessary.

Answer: First of all, thank you for your constructive comments and suggestions that allowed us to greatly improve the quality of the manuscript. As requested by the reviewer the methodology has been revised by adding a paragraph about study population, inclusion and exclusion criteria within methodology section (Please refer line 104-111). Now figure 2 describes about details of Photobiomodulation (LE/P) and quantitative assessment of RR only which makes the readers to understand the concept easily.

Minor language revison required

Answer: Thanks for your valuable comment. The English editing has been done by native English speaker for any typho and grammatical errors.

There are 13 self-citations out of a total of 42 references for the 1st author Mohammad Khursheed Alam, it seems too many self-citations.  I suggest removing the self-citations. 

Answer: Thanks for the valuable input. Around 8 self-citations have been removed. For your reference the removed self-citations were highlighted. The remaining self-citations have been retained as they logically support the rationale of the current manuscript.